# A Deep Learning Approach to Distance Map Generation Applied to Automatic Fiber Diameter Computation from Digital Micrographs

**DOI:** 10.3390/s24175497

**Published:** 2024-08-24

**Authors:** Alain M. Alejo Huarachi, César A. Beltrán Castañón

**Affiliations:** Engineering Department, Pontificia Universidad Católica del Perú, Lima 15088, Peru; cbeltran@pucp.edu.pe

**Keywords:** distance map, regression, deep learning, convolutional neural network, fiber micrograph, synthetic images

## Abstract

Precise measurement of fiber diameter in animal and synthetic textiles is crucial for quality assessment and pricing; however, traditional methods often struggle with accuracy, particularly when fibers are densely packed or overlapping. Current computer vision techniques, while useful, have limitations in addressing these challenges. This paper introduces a novel deep-learning-based method to automatically generate distance maps of fiber micrographs, enabling more accurate fiber segmentation and diameter calculation. Our approach utilizes a modified U-Net architecture, trained on both real and simulated micrographs, to regress distance maps. This allows for the effective separation of individual fibers, even in complex scenarios. The model achieves a mean absolute error (MAE) of 0.1094 and a mean square error (MSE) of 0.0711, demonstrating its effectiveness in accurately measuring fiber diameters. This research highlights the potential of deep learning to revolutionize fiber analysis in the textile industry, offering a more precise and automated solution for quality control and pricing.

## 1. Introduction

The categorization of textile fiber for industrialization and marketing purposes, achieved through diameter measurement, is of great importance in determining the free market price paid to the animal breeder [1,2]; for these reasons, the precise measurement of fiber diameter is crucial in the textile industry, particularly for animal and synthetic fibers, as it directly impacts quality assessment and pricing.

The distinction by observation and touch made by qualified personnel [1] is a method to classify the fiber by its diameter, but since the fibers are very thin (for example, the diameter of alpaca fiber varies between 12 and 28 μm [3]), microscopic analysis is required for greater speed and precision. Thus, equipment has been developed that allows the objective measurement of the characteristics of different types of animal and artificial fiber. The functionalities of this equipment are based on: resistance to the passage of air (Airflow [4]), magnification of observation (projection by microscope [5]), the use of laser beams (SIROLAN-Laserscan [4]), optical processing and image analysis (OFDA [2,6], FIBER-EC [7]) or diffraction of light [8]. Each piece of equipment has advantages and disadvantages, and using them under indicated protocols all measurements have some similarities.

The method based on optical processing and analysis of images has good accuracy and immediacy compared with other methods [6,9]. This method consists of hardware and software equipment, which acquires micrographs of fiber samples and analyzes them automatically. While OFDA measures the average diameter of the fiber in the images, performing pattern recognition processes (although it does not provide details for reasons of confidentiality), FIBER-EC uses well-known computer vision techniques (see Figure 1).

However, automatic fiber diameter measurement in micrographs presents a complicated challenge, partly because the fibers are very close to each other or cross over one another. One approach to address these issues is to employ distance maps [10,11]. The recent achievements in deep learning for computer vision have inspired us to introduce an innovative approach leveraging neural network models. Our method is designed to analyze fiber digital micrographs and generate their corresponding distance maps.

To utilize deep learning models for distance map generation, it is necessary to train these models using numerous fiber images. The effectiveness of these trained models during the testing phase depends greatly on the representativeness of the training set.

The main contributions of this paper include the following:We describe our approach to distance map generation using deep neural network models (which is the first such work to the best of our knowledge).To facilitate the model training, we used sets of real and simulated fiber micrographs, because it is possible to obtain synthetic and realistic digital micrographs.To optimize network learning, three different loss functions are used.We introduce an experimental protocol designed for future reference, ensuring transparent and equitable comparisons.

The rest of the paper is organized as follows. Section 2 reviews some related literature work. In Section 3, the details of our method are presented. The experiment details and results are available in Section 4. Finally, we summarize our findings, provide a discussion concluding remarks, and outline potential future research in Section 5.

## 2. Related Work

Normally, a distance map is the result of the distance transformation algorithm. This idea has been widely used in many fields of research, including computer vision, image analysis, pattern recognition, and so on. This algorithm is generally applied to binary images; and can be used for shape matching and interpolation, skeleton extraction, separation of glued objects, target refinement, etc. [12,13].

For several years, methods for fiber measurement based on morphological operations have been developed for both animal [7,14] and synthetic fibers [10,15]. The fiber measurement process in [10,14,15] begins with acquiring grayscale fiber micrographs. Each micrograph is then segmented, assigning each pixel a value of 0 (background) or 1 (fiber). From here, Baltuano et al. [14] uses the distance map to identify starting points for tracking individual fibers. In contrast, Pourdeyhimi and Dent [15] matches the skeleton to the central pixels of the distance transform, which contain distances to fiber edges, enabling diameter calculation. Ziabari et al. [10] builds upon Pourdeyhimi and Dent [15]’s method but also detects and removes intersections between skeleton lines. Using the resulting image and the distance transform, it calculates fiber diameter with greater precision than the baseline.

Deep-learning-based methods for solving computer vision problems have increased significantly, generating excellent results compared with methods that use classical procedures [16]. In the context of generating distance maps, we identified an FIDT map, introduced by Liang et al. [17], and the DIST technique, introduced by Naylor et al. [11]. While the former proposes distance map prediction for crowd localization, the latter performs nuclei segmentation in histopathology images through distance transform prediction and subsequent U-Net-based processing [18]. Additionally, Huang et al. [12] proposed a two-stage method for left atrium segmentation. It generates the distance map in the first stage to enhance atrium segmentation using U-Net in the second stage. These findings demonstrate the capabilities of neural networks for distance map generation. Therefore, our proposal builds upon these approaches to generate distance maps from fiber micrographs.

On the other hand, we found research that applies deep learning models to recognize the type of animal fiber medulla [19,20]. Additionally, Quispe et al. [20] infers the average fiber diameter based on a classification model.

## 3. Materials and Methods

In this section, we provide details of the deep learning models used and the datasets for training them.

### 3.1. Overall Network Architecture

In our method, we use convolutional neural networks (CNNs) [21], whose output is the expected distance map. Figure 2 is a schematic diagram of the overall architecture of the method. In the figure, the “Distance Map Label” represents the distance map generated with the Distance Transform Method described in Section 3.5, “Input” represents the training image, and “Output” represents the distance map result produced by the network.

### 3.2. CNN Base Architectures

Since we need a CNN to obtain distance maps, we based our approach on the idea of Naylor et al. [11], Huang et al. [12], which modifies the U-Net [18] semantic segmentation model by changing its loss function. This converts the pixel-wise classification into pixel value regression, enabling us to obtain the continuous values required by distance maps. For comparison purposes, we also studied the modification with the SkeletonNet architecture. Next, we provide a brief description of these architectures and their main differences.

U-Net [18] is a CNN designed for biomedical image segmentation. It consists of a symmetric encoder–decoder generator with skip connections, designed for one-to-one mapping problems, making it suitable for mapping fiber micrographs to distance transforms.

The U-Net encoder (downsampling) uses convolutions with 3×3 kernels, ReLU activation function, and 2×2 max pooling for each stage. The U-Net decoder (upsampling) uses 2×2 upconvolutions, concatenation with the skip connection at the same level in the encoder, 3×3 convolutions, and ReLU (see Figure 3).

SkeletonNet [22] is based on U-Net and was designed to extract skeleton pixels from images of various objects. Unlike the plain decoder of U-Net, its decoder design follows the format of the HED architecture [23], which produces four side layers that are merged into a final output layer (see Figure 4).

In addition, before passing through the encoder, the input image is processed by a coordinate convolutional layer [24]. This layer concatenates the input image with two additional channels, *i* and *j*, to map spatial coordinates to the Cartesian space. Residual squeezed blocks [25] are also used in both the encoder and decoder parts, replacing the basic 3×3 convolutions blocks of U-Net. This helps prevent overfitting. In the decoder, the output of these blocks is passed to a channel squeeze and spatial excitation block [26], which maps its input to the spatial location (x,y).

### 3.3. CNN Framework

Our goal is to find a regression model *f* that allows us to predict a distance map *B* from an unseen fiber micrograph *A*. Here, (A,B) is a tuple of a micrograph A∈A and its annotation B∈B, where A is the RGB image space $A=R^{n\times p\times 3}$ and B is the annotation space $B=R^{n\times p}$. The model *f* is found by minimizing the designated loss function (see Section 3.4). Finally, we denote ${\left(A_{l},B_{l}\right)}_{l\in [1,N]}$ as the dataset with annotations, where *N* is the size of the dataset.

### 3.4. Loss Function

To obtain the distance values from the fibers to their edges, and, as proposed by [11], to address the issue of unique segmentation in cases of proximity or overlap, we propose predicting the distance map from fiber micrographs. This approach also allows us to segment the fibers, infer their contours, and subsequently differentiate individual fibers.

We define $B_{D}=\left\{B_{D};B_{D}=Dist_{Euclidean}\left(B\right),B\in B\right\}$ as the ground truth or distance map space. $\hat{B_{D}}=f\left(A\right)$ is the distance map prediction from the input micrograph *A*. Therefore, in the proposed architecture, we use the L1 Loss, L2 Loss, and Smooth L1 Loss (proposed in Fast R-CNN by [27]) criteria. The loss functions are defined in Equations (Equation 1)–(Equation 4).

L1 Loss:(1)$$loss\left(B_{D,l},\hat{B_{D,l}}\right)=\frac{1}{np}\sum_{i,j}\left(B_{D,l}\left[i,j\right]-\hat{B_{D,l}}\left[i,j\right]\right)$$

L2 Loss:(2)$$loss\left(B_{D,l},\hat{B_{D,l}}\right)=\frac{1}{np}\sum_{i,j}{\left(B_{D,l}\left[i,j\right]-\hat{B_{D,l}}\left[i,j\right]\right)}^{2}$$

Smooth L1 Loss:(3)$$loss\left(B_{D,l},\hat{B_{D,l}}\right)=\frac{1}{np}\sum_{i,j}smooth_{L_{1}}\left(B_{D,l}\left[i,j\right]-\hat{B_{D,l}}\left[i,j\right]\right)$$
where
(4)$$smooth_{L_{1}}\left(x\right)=\left\{\begin{array}{c} 0.5x^{2}\;si\;\left|x\right|<1\\ |x|-0.5\;de\;otra\;forma. \end{array}\right.$$

### 3.5. Distance Map Label

The *distance map* is a grayscale image composed of pixels with values of 0 and greater than 0 corresponding to the background and objects, respectively. Each pixel different from 0 has a value equal to the distance to the nearest background pixel [28].

There are several ways to measure the distance between pixels. Here, we used the *Euclidean distance* metric [29,30], which is the straight line distance between two pixels. It is a more realistic metric because it preserves isotropy and is defined in Equation (Equation 5).
(5)$$Dist_{Euclidean}=\sqrt{{\left(x_{1}-x_{2}\right)}^{2}+{\left(y_{1}-y_{2}\right)}^{2}}$$
where $\left(x_{1},y_{1}\right)$ is the position of the pixel with a value greater than 0, and $\left(x_{2},y_{2}\right)$ is the position of the closest background pixel.

To obtain the distance map labels, we first generate binary images from synthetic micrographs (see Figure 5) and then apply the distance transformation algorithm using Euclidean distance.

### 3.6. Datasets

For training the CNN models, we utilize both authentic fiber micrographs and synthetic micrographs due to the scarcity of authentic images and the need for labeling.

#### 3.6.1. Real Micrographs

These images were obtained specifically for our research from OFDA software screenshots; see Figure 6.

#### 3.6.2. Synthetic Micrographs

We developed a computer program to create a set of images that simulate real fiber micrographs acquired by optical processing and image analysis equipment.

To generate synthetic micrographs that resemble real ones, we based their generation on a geometric model, as illustrated in Figure 7.

We used the μ randomness method described by Pourdeyhimi et al. [31], Abdel-Ghani and Davies [32] to generate a network of continuous filaments. Under this scheme, a line with a specific thickness is defined by a perpendicular distance *d* from a fixed reference point *O*, located in the center of the image, and the angular position of the perpendicular α. The distance *d* is limited to the diagonal of the image.

However, fibers often exhibit curvature or are curled; so, for simplicity, a sinusoid is assumed to be sufficient to describe these curvatures. A sinusoid is convenient because it can be easily represented by its length and amplitude. While the length depends on the line size generated by μ randomness, the amplitude is a random value. The wave is then rotated by the angle α.

Each individual fiber is assigned a different grayscale color, and random Gaussian noise is added to each pixel. Noise is also added to the entire micrograph, following the algorithm described by [11].

The result is three-channel RGB color images. Therefore, a micrograph of synthetic fibers is created from a random quantity and diameters. Each fiber is drawn based on the length, position, and orientation of the straight line generated by μ-randomness, with the length and with a random amplitude. The sinusoid representing the fiber’s curvature is created using the length and a random amplitude. Then, each point of the sinusoid is translated according to the position and orientation. Samples of these synthetic micrographs can be seen in Figure 8.

## 4. Experimental Results

In this section, we present an overview of the evaluation protocol, experiments, implementation, and computation time involved in our work. We also provide detailed information on each of these aspects.

### 4.1. Evaluation Protocol

In this protocol, the micrographs, both real and synthetic, are organized into four distinct groups (see Table 1).

In our research, we introduce an assessment protocol (see Figure 9) that outlines the selection criteria for micrographs designated for training and testing purposes:The micrographs from the first and third groups (G1 and G3) are used for training and cross-validation.The second and fourth groups (G2 and G4) are used for testing.

### 4.2. Experiments

The method proposed in this paper is based on the network architecture shown in Figure 2, and the distance map labels are generated using the method introduced in Section 3.5. Based on this, in this section, we conducted experiments that primarily explored the optimization process through three loss functions (see Section 3.4), for each loss function, we trained models based on U-Net and SkeletonNet (see Section 3.2) using Training Subsets G1 and G3 (see Figure 9). For each experiment, we tested the trained models on Testing Subsets G2 and G4 (see Figure 9), and Mean Absolute Error (MAE) and Mean Squared Error (MSE) are reported as the evaluation metrics. The results are summarized in Table 2 and Table 3.

Table 2 and Table 3 show the three optional loss functions, L1 Loss, L2 Loss, and Smooth L1 Loss, used to optimize the training of the two models.

We can see that the model that obtains the best distance map in the real testing subset is U-Net Regression when using L1 Loss with an MAE of 0.1094 and when using L2 Loss with an MSE of 0.0711. The worst results were obtained by SkeletonNet Regression with L2 Loss, although all results are acceptable. Figure 10 and Figure 11 show distance maps of testing in the real subset using the three loss functions and the two models.

On the other hand, the same model (U-Net Regression) compared with the real testing subset obtains the best distance map on the synthetic testing subset. However, this time, it achieves this when using Smooth L1 Loss with an MAE of 0.2139 and when using L2 Loss with an MSE of 0.1229. Again, SkeletonNet Regression obtains the worst result when using L2 Loss. In this case as well, all results are similar and acceptable. Figure 12 and Figure 13 show distance map results of the two models on the synthetic image testing subset using the three loss functions.

### 4.3. Implementation

The experiments were conducted on an Ubuntu machine with Intel Core i9-9920X CPU @3.5 GHZ with 128 GB memory and NVIDIA GeForce RTX 2080 Ti (11 GB) GPU.

The micrographs simulation was performed using a Python program, along with the OpenCV and Pillow libraries. The images can contain from 1 to 15 fibers, each with a random diameter ranging from three to nine pixels.

In addition to using synthetic images, data augmentation [33] was employed using random transformations such as horizontal flip, vertical flip, rotation, Gaussian blur, posterization, and autocontrast. This allows us to increase our dataset size up to $\mathrm{Initial}\_\mathrm{Data}\_\mathrm{Size}\times 2^{T}$, where *T* is the number of transformations, depending on the number of training epochs needed for model convergence.

The models were implemented using Python and the PyTorch framework, based on code from third parties detailed in Table 4. The code and datasets are available on a public repository (Code, and the datasets can be downloaded from https://github.com/alainno/dmnet, accessed on 23 June 2024).

The models were trained from scratch with a batch size of four and an image resolution of 189×189, matching the size of the real OFDA screenshot micrograph. Additionally, the following training configuration was used: optimizer: Adam [34]; initial learning rate: $10^{-3}$, reduced after 10 epochs; maximum epochs: 500; and early stopping when the validation error has not decreased for the last 15 epochs.

**Table 4 sensors-24-05497-t004:** Base codes used in our experiments.

CNN Architecture	Author	Codes Developed By
U-Net	[18]	[35]
SkeletonNet	[22]	[35,36,37,38]

### 4.4. Computational Time

The results in Table 5 demonstrate that the U-Net Regression model has the fastest regression speed. This is primarily due to U-Net having fewer convolutional layers than SkeletonNet. With these results, the computational times achieved could be suitable for real-time videos of up to 12 frames per second. Therefore, it is more than sufficient for improving manual human fiber measurement.

## 5. Discussion and Conclusions

This research aims to improve the analysis of animal and synthetic fiber diameter measurement in micrographs using deep learning, in contrast to [7,10,14], which employs classical computer vision techniques. We focus on the generation of distance maps, a task that has received limited attention within deep learning research, unlike [20], which utilizes a different dataset and a basic classification model. Additionally, we propose a well-defined evaluation protocol that can be replicated and used for comparison in future research seeking to validate their methodology.

Our research objective was to demonstrate how to utilize deep learning strategies to obtain distance maps from fiber micrographs, which contain information representing fiber diameters, rather than developing entirely new architectures. However, our experiments reveal that these strategies cannot be implemented trivially, as they require training on a large number of images to be effective. To address this challenge, we present a deep learning training strategy that employs both real and simulated micrographs. The simulation, based on the μ randomness method with curly filaments, allows us to generate diverse training and testing data with random features. Both types of images were used to train two modified architectures for segmentation and skeletonization, U-Net and SkeletonNet, respectively, adapted for generating distance maps by shifting the objective from classification to regression. These models were evaluated on both real and simulated micrographs.

Regarding the results, both models (U-Net Regression and SkeletonNet Regression) achieve acceptably low errors using different loss functions on both testing subsets. Furthermore, in general, the MSE is lower than the MAE, indicating that the obtained distance maps exhibit a low number of outliers. For a comparative analysis of the error between the test subsets (real and synthetic), the Normalized Root Mean Square Error (NRMSE) is recommended. This metric facilitates comparison given the distinct nature of the two image subsets.

However, the error increases slightly with the SkeletonNet Regression model on both testing subsets and across both error metrics. This could be attributed to the increased complexity of its architecture, which introduces new layers to the original U-Net, specifically the CS-SE block and the dilation layer. These layers focus on the skeleton lines while potentially neglecting other distance map pixels.

Given the existence of various other CNN architectures, it is recommended to perform further comparisons, for example, in terms of cost and speed, with lighter models such as MobileNet, utilizing this architecture as the encoder component of U-Net. In terms of effectiveness, comparisons with Vision Transformers employing Transfer Learning could be made, as these models are trained on millions of images, potentially improving upon the U-Net regression results.

Additionally, the errors on the synthetic testing subset are slightly higher than those on the real testing subset. This suggests a need to enhance the synthetic micrographs with features that more closely resemble real images. Also, it is suggested to experiment with model training using only synthetic data and only real data and measure their impact on the results. It is also recommended to apply OOD (out of distribution) generalization algorithms, such as Invariant Risk Minimization or Adversarial Invariant Learning. This could improve performance, robustness, and accuracy for micrographs under varying lighting conditions, the presence of foreign objects, or unusual backgrounds that were not part of the training data.

On the other hand, the proposed methodology is easy to implement, effective (with similar training and test data), and fast (training takes an average of one hour, while testing requires 0.077 seconds per image). We recommend this method as an alternative for analyzing animal fiber diameter.

In conclusion, our proposed method for generating distance maps from fiber micrographs using deep learning demonstrates significant potential for automating and improving fiber analysis in the textile industry. The results achieved in this study provide a solid foundation for further research and development in this area.

## Figures and Tables

**Figure 1 sensors-24-05497-f001:**
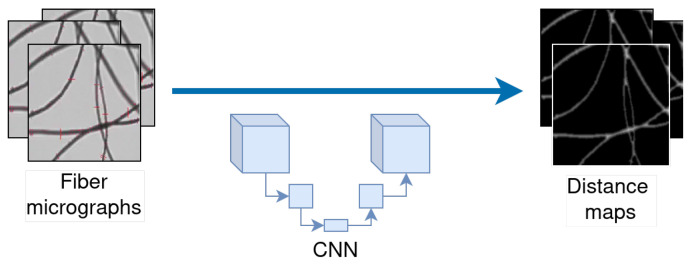
Illustration of a deep neural network applied to distance map computation (Section 3.5).

**Figure 2 sensors-24-05497-f002:**
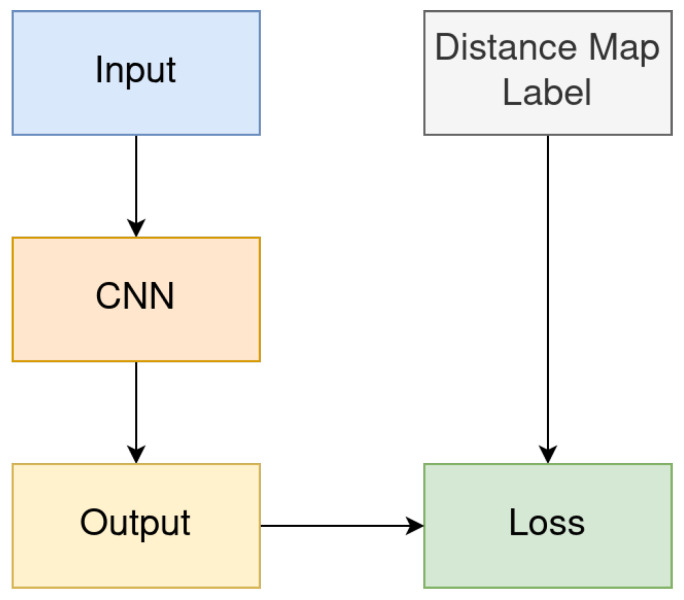
Schematic diagram of the overall network architecture.

**Figure 3 sensors-24-05497-f003:**
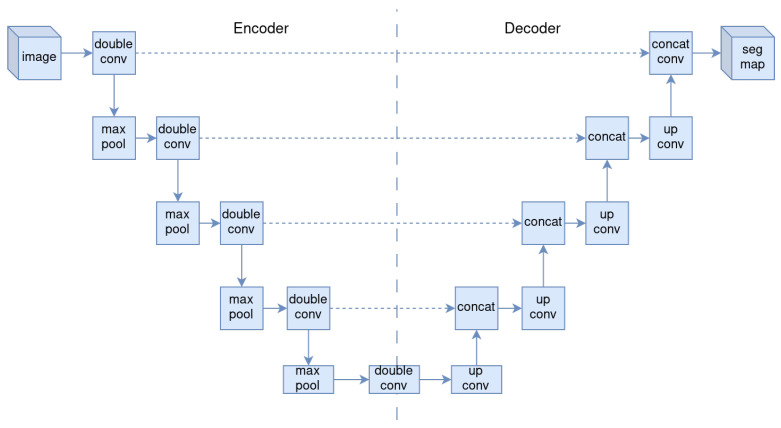
Illustration of U-Net architecture.

**Figure 4 sensors-24-05497-f004:**
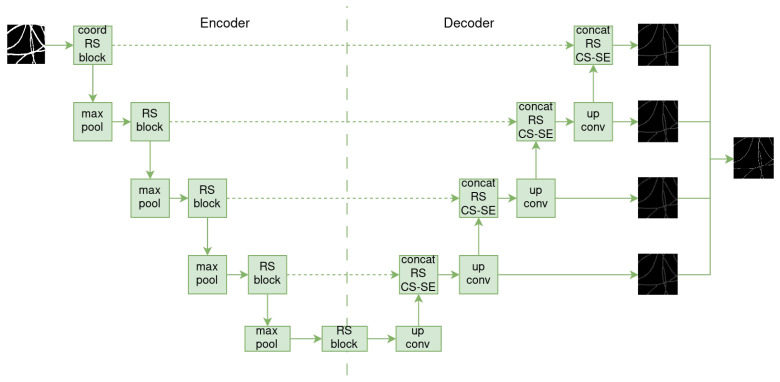
Illustration of SkeletonNet architecture.

**Figure 5 sensors-24-05497-f005:**
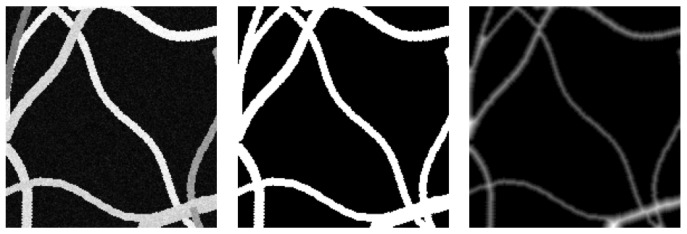
Distance map label steps creation: Left: source synthetic micrograph; middle: binary image from source; right: distance map obtained from binary image with Euclidean distance transform algorithm (Equation (Equation 5)).

**Figure 6 sensors-24-05497-f006:**
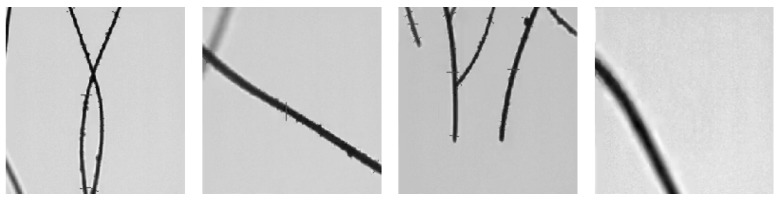
Samples of real OFDA fiber micrographs.

**Figure 7 sensors-24-05497-f007:**
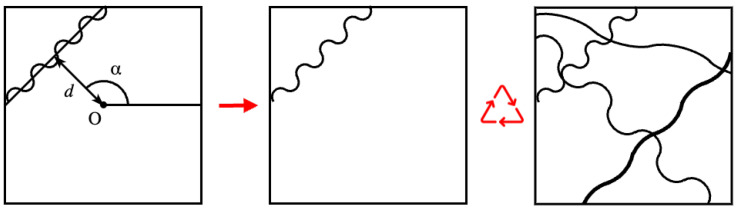
Simulation of fiber micrograph by geometric model: a representation of the μ-randomness method with sinusoid (**left**); the result with one fiber (**middle**); a synthetic micrograph sample with four fibers (**right**).

**Figure 8 sensors-24-05497-f008:**
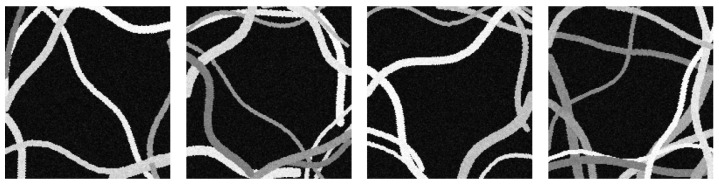
Samples of synthetic micrographs with random features (number of fibers, thicknesses, curvatures, colors, and noise).

**Figure 9 sensors-24-05497-f009:**
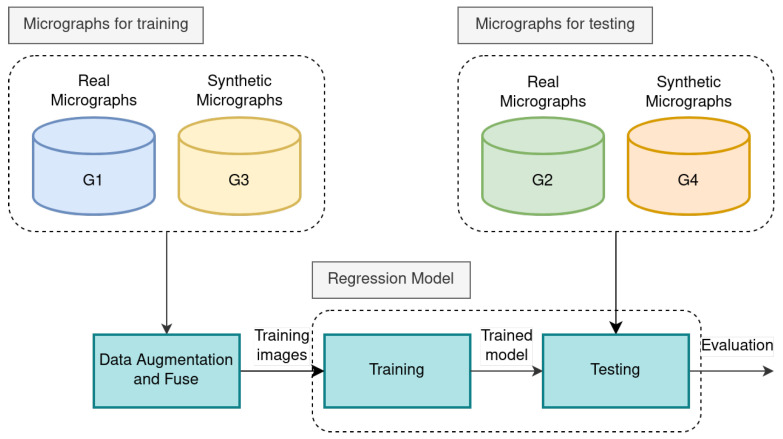
Proposed experimental protocol: In the training stage, the real and synthetic micrographs for training are augmented and fused. In the testing stage, the real and synthetic micrographs are used.

**Figure 10 sensors-24-05497-f010:**
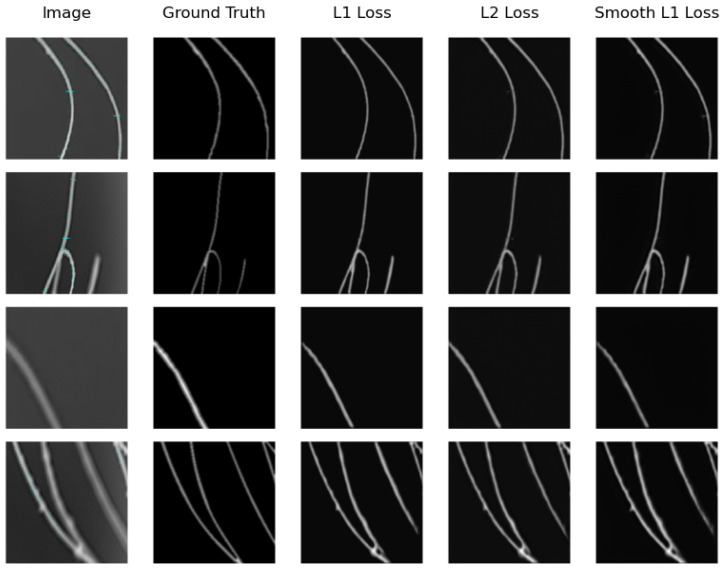
Samples of distance map results of the U-Net Regression using different loss functions on the real image testing subset.

**Figure 11 sensors-24-05497-f011:**
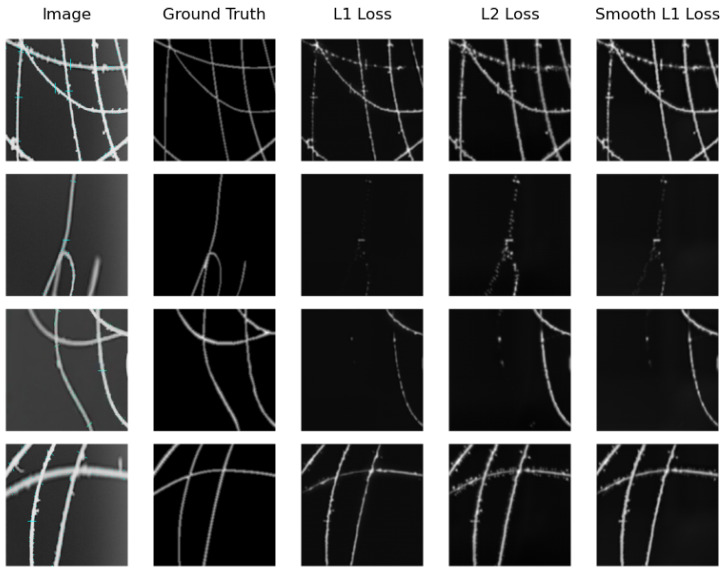
Samples of distance map results of the SkeletonNet Regression using different loss functions on the real image testing subset.

**Figure 12 sensors-24-05497-f012:**
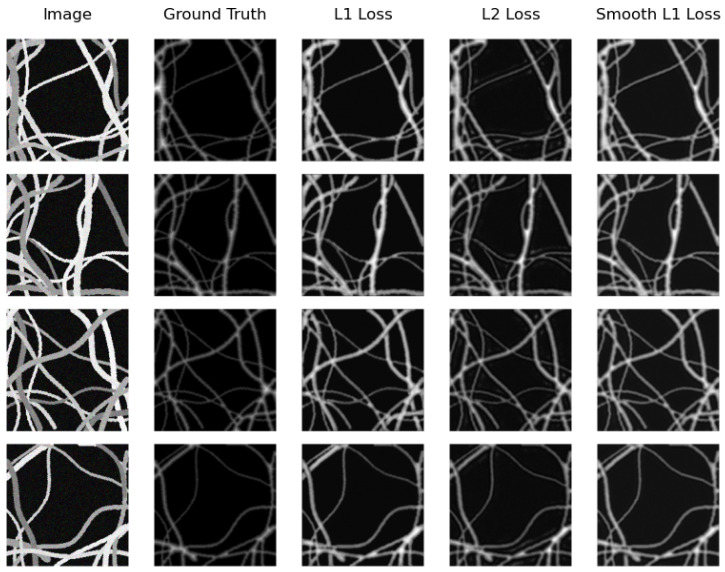
Samples of distance map results of the U-Net Regression using different loss functions on the synthetic testing subset.

**Figure 13 sensors-24-05497-f013:**
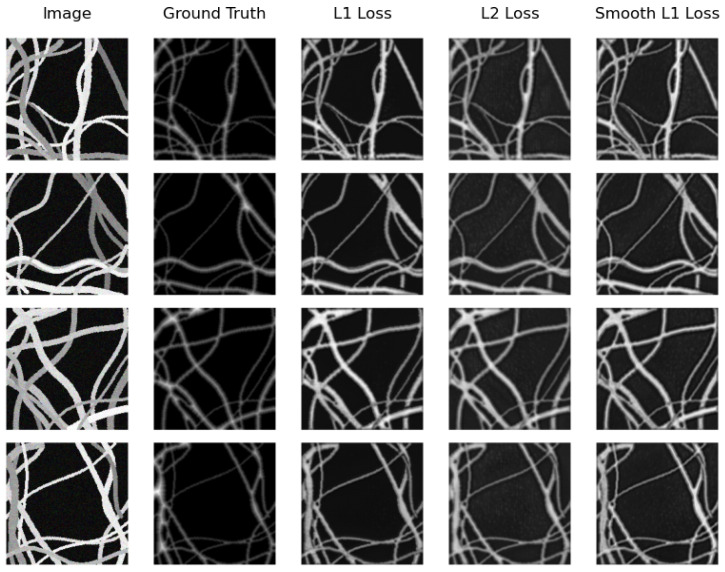
Samples of distance map results of the SkeletonNet Regression using different loss functions on the synthetic testing subset.

**Table 1 sensors-24-05497-t001:** Groups of micrographs defined by experimental protocol.

Group	Purpose	Database *	Images
G1	Training	1	43
G2	Testing	1	11
G3	Training	2	43
G4	Testing	2	11

* 1: Real micrographs, 2: Synthetic micrographs.

**Table 2 sensors-24-05497-t002:** MAE and MSE on the real testing subset.

Model	Loss Function	MAE	MSE
U-Net Regression	L1 Loss	0.1094	0.0882
	L2 Loss	0.1319	0.0711
	Smooth L1 Loss	0.1137	0.0759
SkeletonNet Regression	L1 Loss	0.2041	0.2880
	L2 Loss	0.3087	0.2473
	Smooth L1 Loss	0.2592	0.2519

**Table 3 sensors-24-05497-t003:** MAE and MSE on synthetic testing subset.

Model	Loss Function	MAE	MSE
U-Net Regression	L1 Loss	0.2268	0.1958
	L2 Loss	0.2467	0.1229
	Smooth L1 Loss	0.2139	0.1726
SkeletonNet Regression	L1 Loss	0.2755	0.2717
	L2 Loss	0.3223	0.1892
	Smooth L1 Loss	0.3093	0.2851

**Table 5 sensors-24-05497-t005:** Regression time.

Model	Time per Image [s]	Images per Second
U-Net Regression	0.077	12.987
SkeletonNet Regression	0.094	10.638

## Data Availability

Data are contained within the article.

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
