# Peer review of "A Deep Learning Approach to Distance Map Generation Applied to Automatic Fiber Diameter Computation from Digital Micrographs"

_sensors, 2024, doi:10.3390/s24175497_

Round 1
Reviewer 1 Report
Comments and Suggestions for Authors
Paper proposes a unique approach using deep learning models, specifically U-Net and SkeletonNet. However, the comparative analysis between these models seems insufficiently robust. The performance metrics (MAE and MSE) show significant differences, especially the poor performance of SkeletonNet in comparison to U-Net. The authors should consider a more thorough comparative analysis, potentially including more models or a deeper examination of why SkeletonNet underperforms significantly.
The distinction between real and synthetic micrographs is noted, but the validation of synthetic data's realism and its impact on model training is unclear. Section 3.6.1 and 3.6.2 lacks detail on how these synthetic images are validated against real-world conditions, which could affect the model's practical applicability.
The network architectures in Section 3.2 could benefit from more detailed explanations or graphical illustrations to aid understanding.
Comments on the Quality of English LanguageE.g., the use of "has" instead of "have" in the section discussing CNN base architectures, and punctuation errors scattered throughout the text.
Author Response
Thank you very much for taking the time to review this manuscript. Please find the detailed responses below and the corresponding revisions/corrections highlighted/in track changes in the re-submitted files.
Comments 1: Paper proposes a unique approach using deep learning models, specifically U-Net and SkeletonNet. However, the comparative analysis between these models seems insufficiently robust. The performance metrics (MAE and MSE) show significant differences, especially the poor performance of SkeletonNet in comparison to U-Net. The authors should consider a more thorough comparative analysis, potentially including more models or a deeper examination of why SkeletonNet underperforms significantly.
Response 1: In paragraph 3 of the Discussion and Conclusions Section (Page 12, Line 273), an analysis was added to explain why SkeletonNet exhibits higher errors compared to U-Net: “This could be attributed to the increased complexity of its architecture, which introduces new layers to the original U-Net, specifically the CS-SE block and the dilation layer. These layers focus on the skeleton lines while potentially neglecting other distance map pixels.”
Comments 2: The distinction between real and synthetic micrographs is noted, but the validation of synthetic data's realism and its impact on model training is unclear.
Response 2: Section 3.6.2 (Page 6, Line 160) describes the simulation of synthetic data, incorporating characteristics found in real images, such as the randomness of shapes and orientations, fiber curvature, and noise. Since the primary objective was to minimize errors in tests with real data, the study did not include results from a model trained exclusively with synthetic data; hence, the impact could not be quantified. However, this approach has been recommended for future investigation in Section 5 (Page 12, Line 285): “Also, it is suggested to experiment with model training using only synthetic data and only real data, and measure its impact on the results”.
Comments 3: Section 3.6.1 and 3.6.2 lacks detail on how these synthetic images are validated against real-world conditions, which could affect the model's practical applicability.
Response 3: Section 3.6.1 presents real-world images, while Section 3.6.2 describes synthetic images. Both subsets were merged and used to train two models (see Section 3.2). The performance of these models on real images is detailed in Table 2. Consequently, their applicability is concluded in Section 5 (Page 12, Line 292).
Comments 4: The network architectures in Section 3.2 could benefit from more detailed explanations or graphical illustrations to aid understanding.
Response 4: Agree. We have, accordingly, modified the Section to emphasize this point:
Further details on both architectures, U-Net and SkeletonNet, including their respective layers and blocks, have been included in Section 3.2 (Pages 3 and 4, Lines 107 and 115).
Architecture diagrams for both models are provided in Figures 3 and 4 (Page 4).
Reviewer 2 Report
Comments and Suggestions for Authors
This research introduces a method for generating distance maps of digital samples of animal and synthetic textile fiber micrographs. The approach utilizes deep learning models, the proposed method employs distance map regression to enhance fiber measurement and segmentation.
The models are built upon the U-Net architecture and trained and tested using both real and simulated micrographs.
1- The article needs thorough English review.
2- More details about the CNN model need to be thoroughly explained.
3- The authors should add graphs of the detailed CNN model.
4- The authors should explain the different layers of the CNN model.
Comments on the Quality of English Language
The article requires a comprehensive review for English language and grammar.
Author Response
Thank you very much for taking the time to review this manuscript. Please find the detailed responses below and the corresponding revisions/corrections highlighted/in track changes in the re-submitted files.
Comments 1: The article needs thorough English review.
Response 1: We have reviewed the English text and grammar, making the necessary corrections.
Comments 2: More details about the CNN model need to be thoroughly explained.
Response 2: More details have been added for both models, U-Net and SkeletonNet, in Section 3.2 (Pages 3 and 4, Lines 107 and 115).
Comments 3: The authors should add graphs of the detailed CNN model.
Response 3: The architecture diagrams for both models, U-Net and SkeletonNet, have been added. See Figures 3 and 4 (Page 4).
Comments 4: The authors should explain the different layers of the CNN model.
Response 4: We have explained the layers and blocks that comprise each architecture, U-Net and SkeletonNet, in Section 3.2 (Pages 3 and 4, Lines 107 and 115).
Reviewer 3 Report
Comments and Suggestions for Authors
This paper considers an interesting problem of automatic fiber diameter computation. This is a practical setting and the method is effecitve. The overall content is interesting but the writing not only English need major improvements to make it better. Detailed explainations left for later. For contents, there are some comments:
1.Better use normalized RMSE throughout the paper. As different applications require different precisions, using the normalized version making readers easy to comprehend.
2.Abalations on the synthetic dataset's effectiveness.
3.There are numerous methods can be compared with, especially considering the real scenario's requirement for expenses and speed.
For example,
[1]MobileNet
[2]Linear Vision Transformers
[3]Vision Transformers
There are many more methods should be cited and compared with.
4.How does this method perform in ood scenario where test set is distinct from the training set. For example, train on synthetic fiber and test on alpaca fiber?
There are some works recommended reading and discussing in the paper to enhance its deployability.
[1]Invariant Risk Minimization
[2]OoD-Bench
[3]DecAug AAAI
[4]Adversarial Invariant Learning
[5]Certifiable Out-of-Distribution Generalization
Comments on the Quality of English Language
There are some suggestions for restructing the paper. The English also need major editing.
[1]Put more figures in the intro to illustrate the problem
[2]Shortly discuss related works in the intro
[3]Release the source code in later versions to make the work accessible.
Author Response
Thank you very much for taking the time to review this manuscript. Please find the detailed responses below and the corresponding revisions/corrections highlighted/in track changes in the re-submitted files.
Comments 1: Better use normalized RMSE throughout the paper. As different applications require different precisions, using the normalized version making readers easy to comprehend.
Response 1: This point has been recommended for future investigation in Section 5 (Page 12, Line 268): “For a comparative analysis of the error between the test subsets (real and synthetic), the Normalized Root Mean Square Error (NRMSE) is recommended. This metric facilitates comparison given the distinct nature of the two image subsets.”
Comments 2: Abalations on the synthetic dataset's effectiveness.
Response 2: Since the primary objective was to minimize errors in tests with real data, the study did not include results from a model trained exclusively with synthetic data; hence, the impact could not be quantified. However, this approach has been recommended for future investigation in Section 5 (Page 12, Line 285): “Also, it is suggested to experiment with model training using only synthetic data and only real data, and measure its impact on the results”.
Comments 3: There are numerous methods can be compared with, especially considering the real scenario's requirement for expenses and speed.
For example,
[1]MobileNet
[2]Linear Vision Transformers
[3]Vision Transformers
There are many more methods should be cited and compared with.
Response 3: We agree with conducting comparisons with other models. Since this requires significant modifications, we recommend this for future research, as described in Section 5 (Page 12, Line 277).: “Given the existence of various other CNN architectures, it is recommended to perform further comparisons, for example, in terms of cost and speed, with lighter models such as MobileNet, utilizing this architecture as the encoder component of U-Net. In terms of effectiveness, comparisons with Vision Transformers employing Transfer Learning could be made, as these models are trained on millions of images, potentially improving upon the U-Net regression results.”
Comments 4: How does this method perform in ood scenario where test set is distinct from the training set. For example, train on synthetic fiber and test on alpaca fiber?
There are some works recommended reading and discussing in the paper to enhance its deployability.
[1]Invariant Risk Minimization
[2]OoD-Bench
[3]DecAug AAAI
[4]Adversarial Invariant Learning
[5]Certifiable Out-of-Distribution Generalization
Response 4: We agree with implementing these algorithms and have included this recommendation in Section 5 (Page 12, Line 287): “It is also recommended to apply OOD (out of distribution) generalization algorithms, such as Invariant Risk Minimization or Adversarial Invariant Learning. This could improve performance, robustness, and accuracy for micrographs under varying lighting conditions, the presence of foreign objects, or unusual backgrounds that were not part of the training data.”
Comments 5: There are some suggestions for restructing the paper. The English also need major editing.
[1]Put more figures in the intro to illustrate the problem
[2]Shortly discuss related works in the intro
[3]Release the source code in later versions to make the work accessible.
Response 5:
- We have reviewed the English text and grammar, making the necessary corrections.
- Figure 1 of the illustration of a Deep Neural Network applied to Distance Map computation was added to Page 2.
- Section 2 (Page 2, Line 58) presents the related works..
- The source code has been updated with new instructions to replicate the experiments. (https://github.com/alainno/dmnet)